# Prospective Life Cycle Assessment and Cost Analysis of Novel Electrochemical Struvite Recovery in a U.S. Wastewater Treatment Plant

Karla G. Morrissey [1,*], Leah English [2], Greg Thoma [1] and Jennie Popp [2]

1   Department of Chemical Engineering, University of Arkansas, Fayetteville, AR 72701, USA
2   Department of Agricultural Economics & Agribusiness, University of Arkansas, Fayetteville, AR 72701, USA
*   Correspondence: kgm002@uark.edu

**Abstract:** Nutrient recovery in domestic wastewater treatment has increasingly become an important area of study as the supply of non-renewable phosphorus decreases. Recent bench-scale trials indicate that co-generation of struvite and hydrogen using electrochemical methods may offer an alternative to existing recovery options utilized by municipal wastewater treatment facilities. However, implementation has yet to be explored at plant-scale. In the development of novel nutrient recovery processes, both economic and environmental assessments are necessary to guide research and their design. The aim of this study was to conduct a prospective life cycle assessment and cost analysis of a new electrochemical struvite recovery technology that utilizes a sacrificial magnesium anode to precipitate struvite and generate hydrogen gas. This technology was modeled using process simulation software GPS-X and CapdetWorks assuming its integration in a full-scale existing wastewater treatment plant with and without anaerobic digestion. Struvite recoveries of 18–33% were achieved when anaerobic digestion was included, with a break-even price of $6.03/kg struvite and $15.58/kg of hydrogen required to offset increased costs for recovery. Struvite recovery reduced aquatic eutrophication impacts as well as terrestrial acidification impacts. Tradeoffs between benefits from struvite and burdens from electrode manufacturing were found for several impact categories.

**Keywords:** electrochemical struvite precipitation; life cycle assessment; economic analysis; nutrient recovery and recycling; wastewater treatment





## 1. Introduction

Phosphorus is a crucial element necessary to sustain life, and this nonrenewable resource is expected to be depleted as early as in the next 100 years [1]. Municipal wastewater has been targeted as a primary source for recycling phosphorus as nearly 100% of phosphorus consumed by humans is excreted [2]. Phosphorus is typically removed through secondary biological treatment processes. While these methods are effective in meeting most effluent water quality standards, they result in the production of low-value biosolids that are often transported to landfills [3]. While phosphate is commonly used as an input in the production of detergents, processed food and beverage items, and animal feeds, more than 90% of mined phosphate is consumed by the fertilizer industry [4]. Due to its importance in maintaining global food security, a variety of new phosphorus recovery technologies have been developed in recent decades that aim to selectively recover P-containing products/fertilizers from wastewater treatment plants (WWTPs). Additionally, tools such as life cycle assessment (LCA) and cost analysis have been used to determine the environmental and economic feasibilities of these new technologies that recover phosphorus using chemical precipitation of struvite fertilizer, or $MgNH_4PO_4 \cdot 6H_2O$, from liquid streams [5–9] as well as from solid streams (i.e., digested sludge or sludge ash) [10–13]. Studies evaluating technologies that recover nutrients from digested sludge and/or sludge ash have shown that higher P recoveries are achievable with these pathways as opposed to recovery

methods from liquid streams [8,14]. However, tradeoffs exist for some these technologies as chemical and energy requirements are typically higher, reducing environmental and economic benefits achieved from nutrient recovery [8]. LCAs of technologies that recover nutrients from liquid streams (Ostara, Airprex) have shown that chemicals required for Mg dosing for struvite formation such as $MgCl_2$, $Mg(OH)_2$, or MgO compounds may contribute significantly to environmental impacts [8,15]. Additionally, when benchmarked to the current market price for comparable phosphate fertilizers, the use of chemical inputs for struvite recovery are often greater than the value generated through the sale of struvite fertilizers [13,16–18]. Thus, there is a need to continue improving existing nutrient recovery technologies as well as developing new nutrient recovery methods, keeping design for the environment and costs in mind at the early stages of research and development.

Electrocoagulation has been used as an effective method for removing contaminants from wastewater to meet environmental regulations [19]. However, this process has yet to be utilized for the recovery of struvite in a full-scale municipal wastewater treatment setting. Recent lab studies have shown that electrochemical precipitation may be a feasible technology that can produce struvite from source-separated urine [20], synthetic wastewater [21–25], poultry wastewater [24,26], swine wastewater [27] and real municipal wastewater [26]. In electrochemical precipitation, $Mg^{2+}$ ions can be introduced into an aqueous solution by applying a current through a cell with a sacrificial Mg anode, either pure or alloy [21]. Bench-scale experiments have shown that high-quality struvite production is possible with electrochemical precipitation, with some studies reporting struvite yields of 38–54% based on initial phosphate and ammonium ions in synthetic wastewater solutions [22,23]. In addition to producing struvite, this system has also been shown to co-produce hydrogen, thus offering the potential to offset both economic and environmental energy costs [28,29]. Electrochemical precipitation could be advantageous compared to chemical precipitation in that the addition of fewer chemicals may be required for struvite formation. Previous studies show mixed results when comparing the cost of electrochemical versus chemical precipitation methods at bench-scale [20,25,27]. Additional benefits from the hydrogen gas co-product may also prove to be significant. Some studies have discussed the possibility of hydrogen capture, coupled with struvite recovery using a Mg anode [24,28,30], but none have attempted to model and quantify these processes at full-scale. As this technology undergoes further development it is beneficial to assess potential environmental and economic profiles at these early stages of development for future guidance in design. The aim of this study was to conduct a prospective LCA and cost analysis of electrochemical struvite precipitation to determine hotspots in its current environmental and economic performance as well as to compare these results to those of existing nutrient recovery technologies.

Prospective LCA aims to determine environmental impacts of implementing new and emerging technologies that have not reached full maturity but are modeled in a future large-scale system [31]. While a standard methodology does not yet exist for conducting these types of analyses, some studies in the literature have provided some guidelines. These guidelines recommend modeling emerging technologies as they would most likely perform in an industrial-scale system using a combination of simulation, theoretical and experimental data for life cycle inventory (LCI) collection [32]. For this analysis, electrochemical struvite recovery is modeled as it would be implemented in a typical wastewater treatment plant utilizing the common activated sludge treatment process. This technology is modeled in a treatment plant with and without biosolids treatment to capture the effects of implementing struvite recovery in these systems. A combination of lab-scale data, theoretical calculations, and process simulation was used for LCI collection. The wastewater treatment modeling software GPS-X (v.8.1) and CapdetWorks (v.4) (Hydromantis Inc., Hamilton, ON, Canada) are used process simulation modeling and economic estimation, respectively. Life cycle inventories and impact assessment were collected and generated using LCA software SimaPro (v.9.1) (Pre Consultants, Amersfoort, The Netherlands) [33]. The P-Street WWTP in Fort Smith, Arkansas was used as a reference case study to determine potential

environmental and economic benefits and costs of implementing this emerging technology to existing WWTPs.

## 2. Materials and Methods

### 2.1. Goal, Scope and Functional Unit

The goal of this study is to determine environmental and economic profiles of electrochemical struvite technology based on varying struvite recovery rates, optional $H_2$ capture, and the addition of an anaerobic digester to the existing wastewater treatment plant. The scope of the study includes all material and energy requirements for existing wastewater treatment infrastructure, an optional anaerobic digester, electrochemical struvite recovery and hydrogen capture technology.

Life cycle assessment requires selection of a functional unit to serve as the reference flow for each treatment scenario to ensure comparability across scenarios. Some LCAs in the literature use a unit of phosphorus or fertilizer recovered (i.e., 1 kg of phosphorus or $P_2O_5$ recovered) as the functional unit [5,8,14]; however, not all of the scenarios included in this study produce struvite (described further in Section 2.2). Additionally, the service provided by the WWTP is to treat wastewater. Thus, the chosen functional unit for this study was 1 $m^3$ of treated wastewater for each of the wastewater treatment scenarios. A life cycle impact assessment (LCIA) was generated in SimaPro using the IMPACT World+ MidPoint 1.22 characterization method [34,35].

### 2.2. Description of Case Study and Scenarios

The P-Street WWTP located in Fort Smith, Arkansas was selected as a reference case study. This plant was built in 1966 and has since undergone various upgrades as wastewater treatment technology has improved. This plant utilizes contact stabilization to treat wastewater. The plant has a current treatment capacity of 45,425 $m^3$/day (12 MGD) dry weather flow and a peak design flow of 314,189 $m^3$/day (83 MGD). This plant has a pollution load of 189,271 population equivalents (PEs) based on an average per capita $BOD_5$ loading of 60 g $BOD_5$ per day [36]. Data provided by plant managers was used to model and build scenarios in GPS-X and CapdetWorks that reflected the operations and water quality parameters shown for the plant. Table 1 provides influent characteristics for wastewater, both measured values as well as values that were calculated or taken from literature due to lack of monitoring data.

**Table 1.** Monthly average characteristics of influent wastewater.

| Parameter | Unit | Value |
|---|---|---|
| Average Flow | $m^3$/day | 45,425 |
| TSS [1] | $g/m^3$ | 215 |
| cBOD5 [2] | $g/m^3$ | 250 |
| COD [1] | $g/m^3$ | 431 |
| Ammonia N | $g/m^3$ | 11.0 |
| Nitrite N [2] | $g/m^3$ | 0 |
| Nitrate N [2] | $g/m^3$ | 0 |
| TKN [1] | $g/m^3$ | 15.7 |
| Soluble $PO_4$-P [1] | g P/$m^3$ | 4.8 |
| TP | g P/$m^3$ | 6.0 |
| Total Alkalinity [2] | g $CaCO_3$/$m^3$ | 250 |
| Soluble Mg [2] | $g/m^3$ | 50 |
| pH [2] | - | 7.0 |
| DO [2] | g $O_2$/$m^3$ | 0 |
| Liquid Temperature | °C | 20 |

[1] Values (composite variables) calculated by GPS-X based on inputs. [2] Assumed values based on average influent characteristics from literature [37].

The P Street plant in Ft. Smith utilizes activated sludge treatment with the use of a (1) grit chamber, (2) anoxic reactors (bioselectors), (3) contact stabilization basins, (4) a secondary clarifier, (5) sludge dewatering, and (6) chlorine disinfection units. Various design parameters used for each of the units in these models are shown in Table 2. The Carbon Footprint—Carbon, Nitrogen, Phosphorus, pH "mantis3lib library" was used for process modeling in GPS-X. All unit processes listed in Table 2 below were available in the GPS-X modeling software. However, as electrochemical struvite recovery is a novel technology, it does not currently exist as a unit process in the Hydromantis modeling suite. Thus, a "Black Box 1" process was used that allows for user-specified inputs. Its code was modified to model electrochemical struvite precipitation based on the balanced chemical reaction equation shown below

$$Mg^{2+} + NH_4^+ + H_nPO_4^{n-3} + 6H_2O \rightarrow MgNH_4PO_4 \cdot 6H_2O \downarrow + nH^+$$

where n depends on the pH of the solution and can take the value of 0, 1 or 2 [22]. Struvite precipitation was the only chemical reaction modeled in this black box process. The mass coefficients used for each of the constituents involved ($Mg^{2+}$, $NH_4^+$, $PO_4^{3-}$ ions and struvite) are given in the Supplementary Materials (Section S1).

**Table 2.** Design parameters for contact stabilization treatment train at P Street plant.

| Unit Process | Design Parameters | Values |
|---|---|---|
| Grit chamber | Grit production per flow (m$^3$/L) <br> Dry solid content of grit (%) | 20 <br> 98 |
| Anoxic reactors (Bioselectors) | Tank depth (m) <br> Maximum volume (m$^3$) | 4 <br> 2650 [1] |
| Plug-flow contact basins | Number of reactors in series <br> Tank depth (m) <br> Maximum volume (m$^3$) | 3 <br> 4 <br> 6321 |
| Plug-flow stabilization basins | Number of reactors in series <br> Tank depth (m) <br> Maximum volume (m$^3$) | 3 <br> 4 <br> 3142 |
| Secondary clarifiers | Surface area (m$^2$) <br> Tank depth (m) <br> RAS/WAS [3] pumped flow (m$^3$/day) | 2917.2 [2] <br> 3 <br> 22,712 |
| RAS/WAS pumping station | WAS pumped flow (m$^3$/day) | 378.5 |
| Dewatering | Underflow solids (mg/L) <br> Solids removal efficiency (%) | 200,000 <br> 95 |
| Disinfection | Volume (m$^3$) <br> Chlorine dosage (mg/L) | 844 <br> 6.0 |
| Gravity thickener [4] | Surface area (m$^2$) <br> Tank depth (m) <br> Underflow solids (mg/L) <br> Removal efficiency (%) | 154.2 <br> 3 <br> 48,000 <br> 90 |
| Anaerobic digester [4] | Maximum volume (m$^3$) <br> Temperature (°C) | 1500 <br> 35 |
| Electrochemical struvite Reactor [5] | Section S1a–d | - |
| Solids separation [5] | Pumped flow (m$^3$/day) <br> Struvite removal efficiency (%) | 1 <br> 99 |

[1] Four anoxic reactors are modeled in GPS-X each with a maximum volume of 662 m$^3$ to give a total of 2650 m$^3$ of anoxic basin volume. [2] Four secondary clarifiers are utilized when treating average flow of 12 MGD. One clarifier is used to model all clarifiers with a combined surface area of 2917.2 m$^2$. [3] RAS: return activated sludge; WAS: waste activated sludge. [4] Units are present in "A" scenarios that include additional anaerobic digestion step preceded by a thickening step. [5] Units are present in scenarios that include additional electrochemical struvite recovery step followed by a solids separation step.

This study includes 10 scenarios in which a basic schematic of the Ft. Smith plant is used and modified. Table 3 summarizes the scenarios included in this analysis. Scenario B1 is a base case scenario that represents "business-as-usual" operation of the existing wastewater treatment plant in Ft. Smith, AR. Scenarios B2-45 and B2-90 follow the base case schematic with the addition of a struvite recovery step placed after dewatering. Struvite is recovered from the dewatering filtrate using an assumed struvite yield (fraction of theoretical yield) of 45% and 90% for B2-45 and B2-90, respectively. These yields were chosen based on the bench-scale results that have been achieved thus far in the literature as mentioned previously [22,23]; a 45% yield is assumed as a current performance yield of the bench-scale technology. A yield of 90% is also modelled as a theoretical "best case scenario" that could potentially be accomplished in the future. B3-45 and B3-90 include the previous modifications as well as hydrogen capture from electrochemical struvite precipitation.

**Table 3.** Scenarios for Electrochemical Struvite Recovery Analysis.

| Scenario | Description | % Struvite Yield | $H_2$ Capture | Anaerobic Digestor |
|:---:|:---:|:---:|:---:|:---:|
| B1 | Existing treatment scheme in Ft. Smith | 0 | No | No |
| B2-45 | Existing treatment scheme with struvite production | 45 | No | No |
| B2-90 | Existing treatment scheme with struvite production | 90 | No | No |
| B3-45 | Existing treatment scheme with struvite production and $H_2$ capture | 45 | Yes | No |
| B3-90 | Existing treatment scheme with struvite production and $H_2$ capture | 90 | Yes | No |
| A1 | Existing treatment scheme with added anaerobic digestor | 0 | No | Yes |
| A2-45 | Added anaerobic digestor with struvite recovery | 45 | No | Yes |
| A3-90 | Added anaerobic digestor with struvite recovery | 90 | No | Yes |
| A3-45 | Added anaerobic digestor with struvite recovery and $H_2$ capture | 45 | Yes | Yes |
| A3-90 | Added anaerobic digestor with struvite recovery and $H_2$ capture | 90 | Yes | Yes |

Scenario A1 models the existing Ft. Smith treatment plant with the addition of a thickener and anaerobic digestion step. Unlike the rest of the base case scenarios, this scenario provides biogas as a by-product of treatment that is assumed to be combusted in a boiler and provide heat and electricity for the plant. This scenario also includes a ferric chloride dosing step to the stream exiting from the anaerobic digester prior to dewatering to precipitate excess soluble phosphate ions according to standard practice. Scenarios A2-45 and A2-90 include a struvite recovery step with similar struvite yields as previously mentioned for scenarios B2-45 and B2-90. Similarly, scenarios A3-45 and A3-90 include hydrogen capture and compression.

The process flow diagram below shows the existing treatment train in the P Street plant in Ft. Smith for the B1 scenario (Figure 1). For B2 and B3 scenarios, this base case configuration was modified to include a struvite recovery step treating the liquid stream directly after the dewatering step. A solids separation step is placed following struvite recovery prior to recycling of treated liquid back to the headworks. No additional Mg dose is provided in these scenarios as soluble Mg is present in excess (influent value). For the A1 scenario, a gravity thickener and anaerobic digester are added to treat WAS placed downstream of the RAS/WAS pump station and upstream of the dewatering step. Similar to the B2 and B3 scenarios, A2 and A3 scenarios include anaerobic digestion as well as a struvite recovery step and solids separation step after dewatering. A Mg dosage is included in these scenarios in order to provide excess soluble Mg for the struvite precipitation

reaction. The PFDs for the B2/B3 scenarios, the A1 scenario and the A2/A3 scenarios are provided in Figures S1–S3 in the Supplementary Material, respectively).

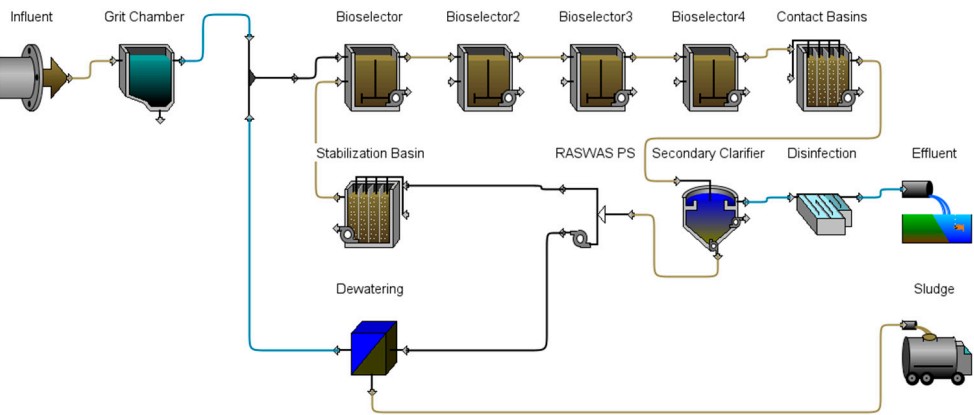

**Figure 1.** GPS-X Process flow diagram for Ft. Smith P Street Plant (B1 scenario).

As mentioned, scenarios A2-45, A2-90, A3-45 and A3-90 require an additional Mg dosing step to provide sufficient soluble Mg for struvite precipitation. As there is no option in GPS-X to provide soluble Mg electrochemically, Mg can only be added to the system chemically as $Mg(OH)_2$ or $MgCl_2$. The Mg dosage is simulated as addition of $Mg(OH)_2$ as this chemical also releases excess hydroxide ions in solution, similar to the electrochemical system. This chemical addition is only provided in this model to simulate struvite precipitation; the life cycle inventory for the electrochemical cell (described in further detail in Section 2.4.3) does not include $Mg(OH)_2$ additions but rather the pure Mg plates required for the cell.

*2.3. System Boundary*

The system boundary for this analysis includes all infrastructure, material and energy inputs that are used for the chosen functional unit of treatment of 1 $m^3$ of municipal wastewater. The system boundary for all B scenarios is shown in Figure 2. All process requirements for Ft. Smith wastewater treatment train were included in the analysis. For B2 and B3 scenarios, co-products including struvite and hydrogen gas were modeled as avoided products that provided environmental credits.

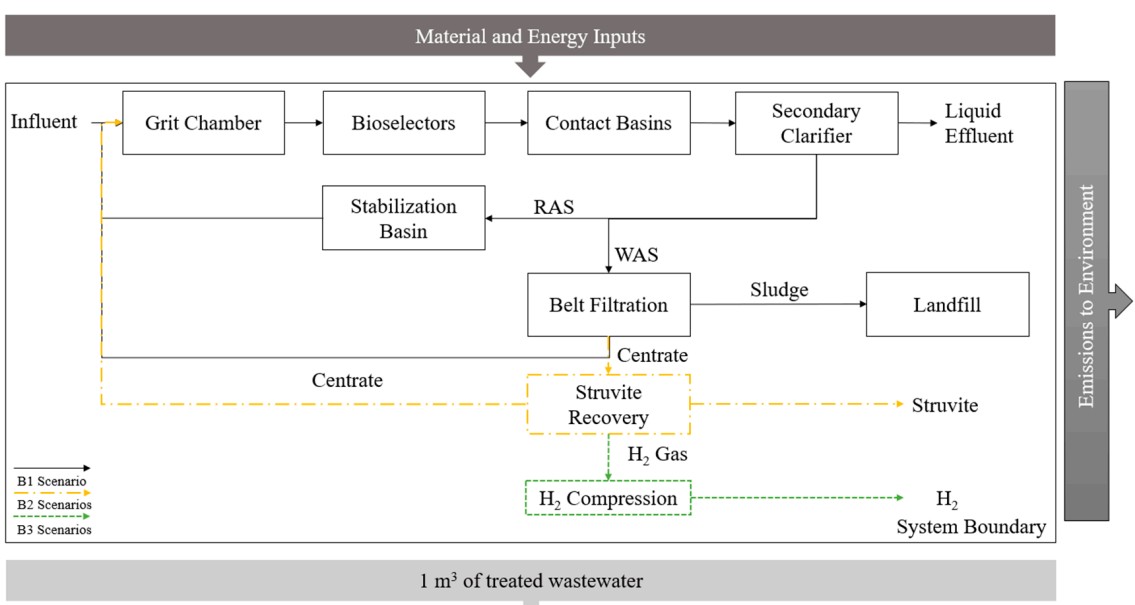

**Figure 2.** System Boundary for Base Case Scenarios with Optional Struvite and $H_2$ Capture.

Figure 3 shows the system boundary for all scenarios that include the addition of an anaerobic digester. Inclusion of an anaerobic digester required an additional separation step using a gravity thickener prior to digestion. In addition to struvite and hydrogen gas, heat and electricity is produced from combustion of biogas produced in these scenarios. Heat and electricity from biogas are modeled as avoided burdens for all A scenarios and provide environmental and economic credits. Process data used for each step is discussed in further detail in Section 2.4.

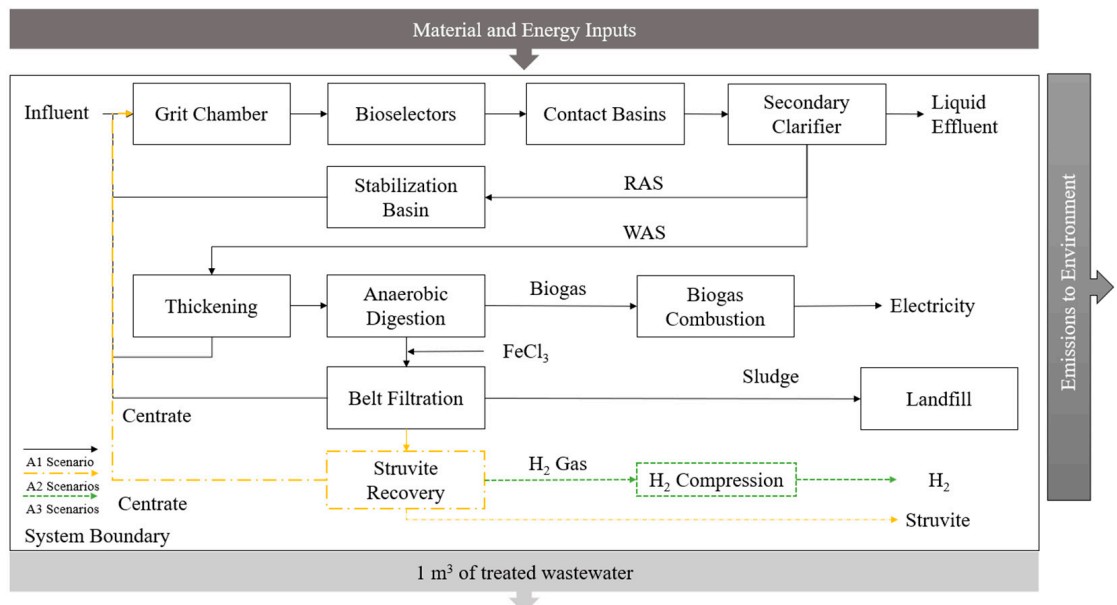

**Figure 3.** System Boundary for Scenarios with Added Anaerobic Digestion.

## 2.4. Life Cycle Inventory

The life cycle inventory collected for this study includes data sourced from GPS-X process simulation modeling, experimental data, and literature data. Derivations for all LCI values used in this study are described in the sections below and provided in Table S1 of the Supplementary Material.

### 2.4.1. Plant Infrastructure, Chemicals, and Sludge Disposal

Many LCAs of nutrient recovery in WWT in the literature exclude construction of plant infrastructure in their life cycle inventory [8,15,38]. However, inclusion of plant infrastructure could be significant as facility construction can contribute up to 80% of the overall impact depending on the treatment process employed [39]. Infrastructure requirements for the wastewater treatment plant were included in the life cycle inventory using values provided by Doka 2021, scaled to the treatment flow rate of the P Street WWTP [40]. Infrastructure requirements for the combined heat and power plant for scenarios with an anaerobic digester were obtained from Whiting & Azapagic and scaled to the combined heat and electricity generated in the scenarios [41]. Chemicals required for treatment were based on chlorine dosage values provided by plant operators. No other chemicals were included in the inventory for all scenarios except for the A1 scenario which included ferric chloride in-line dosage (22.4 kg/h). As the Ft. Smith plant currently landfills all of their dewatered sludge, end-of-life of sludge is modeled as sludge disposed in a sanitary landfill. The inventory for sludge disposal was calculated using the Waste Tool provided by Doka 2021 [42]. This tool calculated all material and energy inputs associated with sludge disposal based on specific sludge characteristics (i.e., water content and elemental composition from GPS-X outputs) as well as site specific information for the Ft. Smith City Landfill (i.e., annual precipitation, evapotranspiration rate).

### 2.4.2. Process Simulation Outputs

Process simulation models were run in GPS-X under steady-state conditions. Information provided by the simulation outputs included energy requirements for aeration, pumping, and heating, and flow and concentration of nutrients for each process stream including solid and liquid effluents. Additionally, amounts and characterization of biogas produced in scenarios with an anaerobic digester were given. The mantis3lib library also provides all Scope 1, 2, and 3 carbon emissions for each unit process as well as the entire plant. These emissions include $CO_2$, $CH_4$, and $N_2O$ emissions from biological treatment processes in units of kg $CO_2$-eq. For this study, only net Scope 1 emissions were used in the life cycle inventory as emissions from electricity and downstream processes were accounted for directly in the LCA model. These Scope 1 emissions were modeled as emissions to air. Nutrients present in the liquid effluent were modeled as emissions to water.

### 2.4.3. Electrochemical Struvite Precipitation Reactor

Materials, energy, and yields for electrochemical struvite precipitation were obtained from experimental results in Kekedy-Nagy et al. [22]. Mg metal requirements for this electrode were estimated as 0.196 kg Mg/kg struvite. The cathode is a stainless-steel electrode that is not consumed during the reaction. A number of LCAs of stainless-steel used in construction/piping of wastewater treatment plants recommend a lifespan of 12.5–15 years for stainless steel [43–45]. For this system, a conservative lifespan of 10 years is assumed for the stainless-steel cathode. Electricity requirements for electrochemical precipitation were 4.6 kWh/kg struvite. A solids separation step was also included and assumed to remove 99% of struvite in the stream directly following struvite precipitation. Struvite produced in scenarios with the electrochemical reactor is modeled to displace the N- and P-equivalents in the market in the form of ammonium nitrate, as N, and phosphorus fertilizer, as $P_2O_5$.

### 2.4.4. $H_2$ Collection and Compression

Few electrochemical struvite recovery studies report hydrogen yields from bench-scale studies [28,29]. Hydrogen gas is generated at the cathode as a result of the current applied and theoretical $H_2$ yields can be found using Faraday's law with known current applied and total mass of Mg that dissociated [29]. However, this calculation has been shown to underestimate yields of hydrogen as hydrogen is also generated at the anode due to the negative difference effect [28,29]. For this study, a conservative estimate of hydrogen using Faraday's law was chosen to determine environmental credits provided by hydrogen gas. An estimated yield of 134 L $H_2$ per kg of struvite is assumed to be generated, sourced from personal communication of ongoing work. Since an industrial-scale electrochemical cell for struvite recovery does not currently exist, it is assumed the cell would most likely resemble that of an alkaline electrolyzer used for water electrolysis. An alkaline electrolyzer is composed of two electrodes immersed in a liquid solution and separated by a gastight diaphragm [46]. Oxygen gas is generated at one electrode, and hydrogen gas is generated at the other. Hydrogen gas purities of >99 vol.% are typically achieved with this type of electrolyzer, negating the need for subsequent purification steps [47]. It is assumed hydrogen gas generated at the cathode will be collected using a similar system. In addition, pH is maintained at a neutral level so that no ammonia gas is formed [48]. Thus, the hydrogen gas collected from electrochemical struvite precipitation is pure and no additional purification steps are required. Hydrogen gas produced in this system is assumed to be used as a fuel source for vehicles utilizing proton exchange membrane (PEM) fuel cells and thus must be compressed. Hydrogen gas will be compressed from 1 atm to 200 atm [49]. Energy requirements for compression were acquired using the isothermal compression equation provided by Granovskii et al. and were estimated at 61.2 kJ/mol $H_2$ assuming ideal gas behavior [50].

### 2.4.5. Biogas Combustion and Emissions

Scenarios with an anaerobic digester produce biogas as a by-product of treatment. The biogas is combusted in a boiler to generate electricity. An efficiency of 75% is assumed for the boiler [51]. Heat recovery and electricity recovery efficiencies were assumed to be 46% and 39%, respectively [41]. Heat and electricity offsets were calculated in GPS-X and included in the inventory (Table S1). Emissions from biogas combustion were calculated based on emission factors provided by Nielsen et al. [52]. Compounds included in emissions to air were sulfur dioxide ($SO_2$), nitrous oxides ($NO_x$), non-methane volatile organic compounds (NMVOCs), and formaldehyde ($CH_2O$). Carbon monoxide and carbon dioxide emissions were not included as these emissions are considered biogenic [53].

### 2.5. Economic Analysis Methods

CapdetWorks software was used to estimate costs for all standard wastewater treatment processes utilized in each scenario. Costs for non-standard processes (i.e., struvite recovery, hydrogen capture, and biogas utilization) were derived from available literature and GPS-X simulation outputs. Capital costs were annualized and combined with operating costs to determine the total project cost for each scenario. Estimated revenues stemming from the sale of struvite fertilizers and hydrogen gas were subtracted from the total costs to determine the expected net costs per $m^3$ of wastewater treated. Capital and input costs for struvite recovery and hydrogen capture were used to estimate break-even prices per kg of struvite and hydrogen gas produced.

### 2.5.1. Modeling Unit Processes in CapdetWorks

CapdetWorks software was used to derive capital and operational costs for each unit process shown previously in Figure 1. Values from the software were also utilized for gravity thickening and anaerobic digestion processes used in each "A" scenario. Plants utilizing anaerobic digestion have been shown to incur additional operational and maintenance costs for the management of struvite scaling [54,55]. Scaling is often averted through the addition of metal salts (e.g., ferric chloride) to bind phosphates released during the digestion process. To account for this, an iron feed system supplying $FeCl_3$ to the digestor effluent was included in cost estimates for the A1 scenario. Controlled struvite precipitation by the electrochemical reactor eliminates the need for scaling control, thus replacing the use of the iron feed system for the A2 or A3 scenarios.

Similar to prior studies [56–58] CapdetWorks layouts were modeled and calibrated based on influent characteristics, unit processes, and system parameters used in the GPS-X simulations described in previous sections. Where possible, cost and pricing data specific to the study region were gathered and updated within the model [59–71]. A complete list of costs and sources can be found in Table S2. Using these input parameters, costs for unit processes were estimated according to the Hydromantis equipment costing database U.S. (2014), coupled with the Hydromantis cost indices within the software to present all values in 2022 $'s [72,73].

### 2.5.2. Costs for Struvite and Hydrogen Recovery

Struvite recovery costs presented in Table 4 were based on specifications and costs reported for the installation and operation of a commercial Pearl 500 struvite reactor manufactured by Ostara, Inc. [74,75]. Chemical and energy costs are represented by magnesium and energy consumed by the struvite reactor (see Section 2.4.3). The price for magnesium is based on the 2021 U.S. spot price for magnesium metal, adjusted for retail markup and inflated to 2022 $'s [63–65].

**Table 4.** Struvite Recovery Cost Parameters (per kg of struvite produced).

| Cost Parameters | Value | Units | Source |
|---|---|---|---|
| Reactor | 1.73 | $ | [74,75] |
| Building Space | 0.43 | $ | [74,75] |
| Labor | 0.006 | h | [74,75] |
| Maintenance | 0.23 | $ | [74,75] |
| Mg (99.9%) | 3.24 | $ | [63–65] |
| Energy | 0.29 | $ | [66] |

Capital and operating costs for hydrogen compression were estimated using methods detailed in Khan et al., (2021) [76]. This methodology involves the use of empirical cost correlations derived from the U.S. Department of Energy's $H_2A$ Hydrogen Delivery Scenario Analysis Model (HDSAM) coupled with expected outputs of hydrogen stemming from the B3 and A3 scenarios [77]. Input parameters for these calculations can be found in Table S3. Energy costs for compression are derived from values estimated using methods discussed in Section 2.4.4. Recovered hydrogen is assumed to be stored on-site in a type II pressure vessel at 200 bar at a cost of $86/kg of $H_2$ stored [78]

2.5.3. Annualization and Functional Unit Conversion

Where necessary, capital costs were annualized by multiplying the total cost by a capital recovery factor (CRF). The CRF was calculated using the following Formula (1). The resulting value represents the annual payment required at a specified interest rate (i) of 8% and planning period (n) of 40 years: [56,57,79]

$$CRF = \frac{i(1+i)n}{\{(1+i)\,n - 1\}} \tag{1}$$

To calculate costs per $m^3$ of treated wastewater, net annualized costs were divided by the average wastewater flow and number of days per year Equation (2):

$$\text{Cost per } m^3 = \frac{(\text{Annualized project costs} + \text{Operating costs}) - \text{Expected Revenues}}{(\text{Avg. daily flow} * 365)} \tag{2}$$

2.5.4. Estimating Potential Revenues and Cost Offsets

Struvite recovery and hydrogen capture each offer an opportunity for potential revenues to be generated through the sale of struvite fertilizer and hydrogen gas. To evaluate this potential, output values for struvite and hydrogen (estimated using methods outlined in Section 2.4) are multiplied by the price of each product. The price of struvite is represented in this primary analysis as the June 2022 price of diammonium phosphate (DAP) fertilizer, $1.15/kg [80]. DAP has a similar phosphate composition to the electrochemically derived struvite and in order to be economically viable, struvite prices would need to be competitive with commercially available phosphate fertilizers. Similarly, since hydrogen fuel production is relatively scarce in the U.S., pricing for hydrogen is based on the gallon gas equivalent (GGE) of gasoline. For gasoline sold at $3.20 a gallon, the equivalent price of hydrogen is $8 per kg [81,82]. Revenues calculated for struvite and hydrogen were then subtracted from the annualized costs to determine if these sales were sufficient to offset costs. Additionally, it is assumed that biogas produced by the anaerobic digestor will be utilized by the plant, partially offsetting energy costs across the A scenarios.

Subsequent break-even analyses were performed to evaluate the price point required to offset annual capital, chemical input, and energy costs used in the production of struvite and hydrogen.

## 3. Results

### 3.1. GPS-X Model Results

Process simulation outputs, primarily flows and characterization of liquid and solid effluents, are given in Table S1. Struvite yields ranged from less than 1 kg/day to 702 kg/day. Hydrogen yields ranged from 1.8 L/day to 94,500 L/day. A summary of struvite yields, hydrogen generated, and heat and electricity generated from biogas (A scenarios) is given in Table 5. Phosphorus recoveries are based on the fraction of influent P captured as struvite.

**Table 5.** By-product yields in wastewater treatment scenarios.

| By-Product | B1 | B2-45 | B2-90 | B3-45 | B3-90 | A1 | A2-45 | A2-90 | A3-45 | A3-90 |
|---|---|---|---|---|---|---|---|---|---|---|
| Struvite (kg/day) | - | 0.013 | 0.026 | 0.013 | 0.026 | 0 | 389 | 702 | 389 | 702 |
| Hydrogen gas (L/day) | - | - | - | 1.81 | 3.59 | - | - | - | 52,330 | 94,500 |
| Electricity, combustion of biogas (kWh) | - | - | - | - | - | 89.1 | 88.3 | 89.7 | 88.3 | 89.7 |
| Heat, combustion of biogas (kWh) | - | - | - | - | - | 124.4 | 123.2 | 125.3 | 123.2 | 125.3 |
| Phosphorus recovered in struvite (%) | - | ~0 | ~0 | ~0 | ~0 | - | 18 | 33 | 18 | 33 |

Struvite recoveries in B scenarios were low due to the low soluble phosphate concentrations in the stream following the dewatering step. Struvite recoveries were higher in A scenarios since the addition of an anaerobic digester allowed for nutrients bound in cell biomass to be released in soluble form. However, release of soluble phosphorus with the addition of the digester resulted in higher amounts of ortho-phosphate circulating in the system, with liquid effluent concentrations in the A1 scenario resulting with double the TP concentration as that of the base case scenario (B1). Ammonia concentration in the liquid effluent for all scenarios remained steady between 0.15–0.19 mg N/L with the lowest concentration achieved by the A1 scenario. In the sludge, TP concentrations vary between 6533–9931 mg P/L, with the lowest TP values achieved by the A2/3-90 scenarios. Sludge volumes for B scenarios ranged between 917–919 kg/h while sludge volumes for A scenarios ranged between 629–729 kg/h. Thus, the anaerobic digester provided an average of a 28% reduction in sludge volumes between the B and A scenarios. Biogas produced across all A scenarios varied between 46.5 to 47.4 $m^3$/h, generating 211–215 kWh of combined heat and electricity. These values offset approximately 22–23% of electricity required for plant operation not including electricity required for electrochemical struvite recovery in those respective scenarios. Heat recovered from biogas was not sufficient to fulfill the digester heating requirement; therefore, natural gas was used to supplement the remaining energy (~8–13 kWh).

### 3.2. Life Cycle Impact Assessment Results

All results were scaled to the functional unit of 1 $m^3$ of treated wastewater. Figure 4 shows the LCIA results for all scenarios across 6 impact categories. The remaining categories are presented in Figure S4.

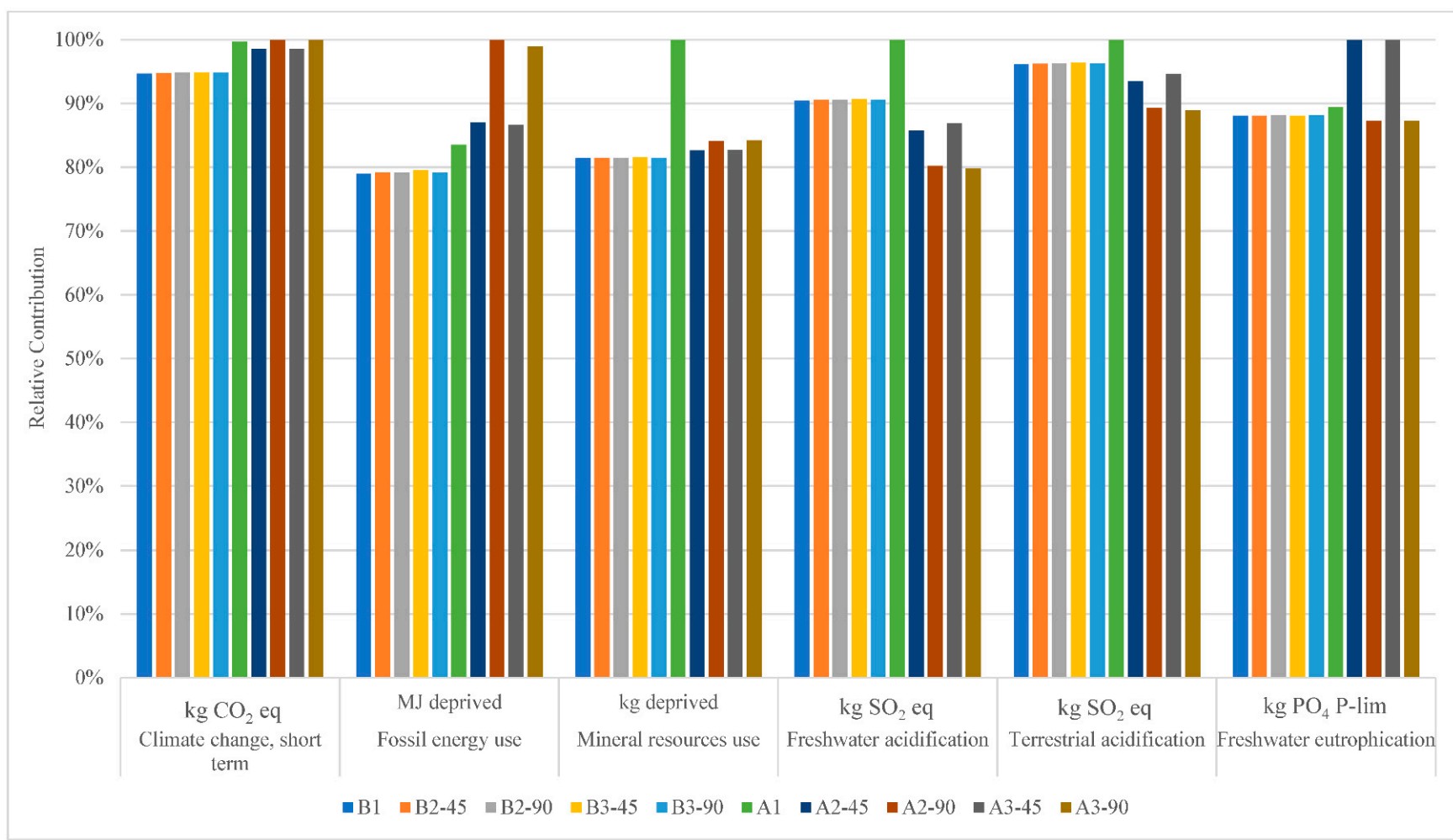

**Figure 4.** LCIA results for wastewater treatment scenarios (IMPACT World+).

LCIA results show that B scenarios had lower relative impacts compared to A scenarios for photochemical oxidant formation, particulate matter formation, short and long-term climate change, fossil energy use, and mineral resources use. For land occupation-biodiversity and human toxicity cancer, this trend is similar with the exception that the A1 scenario has the lowest environmental impacts overall. The B scenarios have higher relative impacts in the categories of water scarcity and human toxicity non-cancer. For freshwater eutrophication, A#-90 scenarios have the lowest impacts and A#-45 have the highest relative impacts. For freshwater ecotoxicity, The A1 scenario has the highest impacts and A2 scenarios have the lowest overall impacts. The A1 scenario has the highest relative impacts across 6 out of the 18 total impact categories and the lowest relative impacts in three categories. Impacts across the B scenarios do not vary significantly as considerably lower struvite yields were produced in these scenarios compared to A scenarios. In contrast, impacts vary significantly across the A scenarios. In general, A#-90 scenarios result with higher relative impacts compared to A#-45 scenarios across all impact categories except water scarcity, freshwater acidification, terrestrial acidification, freshwater eutrophication, marine eutrophication, and ozone layer depletion.

A breakdown of the driving factors for each impact category and scenario is given in Figure 5. Each process that contributes to an environmental impact is shown as a different color. Process contributions as well as overall impacts for each category is referenced on the respective *y*-axis with units given in the *x*-axis. The contribution categories shown in Figure 5 includes treatment of wastewater, electricity, electrode manufacturing, struvite produced, hydrogen produced, sludge transport, treatment facility, chemicals, and landfilling of sludge. Treatment of wastewater includes all emissions to air, water and soil resulting from treating 1 m$^3$ of water (all compounds in liquid and solid effluents from GPS-X and emissions to air). Electricity includes all energy required to operate the treatment facility including the electrochemical struvite precipitation reactor for the scenarios that model struvite recovery. Treatment facility includes the construction of the wastewater treatment plant and of the combined heat and power plant for A scenarios only. All other process contribution categories are as mentioned in the methods section. Negative values in the process contribution graphs show environmental credits as opposed to impacts for that process/category. Process contributions for the remaining categories are provided in Figure S5a,b.

Chemicals did not significantly contribute to LCIA results in most of the scenarios and categories except for the A1 scenario in fossil energy and mineral resources use and ozone layer depletion. Facility construction as well as the treatment process were some of the largest contributors across the impact categories. Electrode manufacturing also contributed significantly to a few impact categories. Credits for struvite and hydrogen were present in nearly all impact categories with the exception of freshwater eutrophication, marine eutrophication, and water scarcity. Further analysis of these process contribution results is given in Sections 3.2.1–3.2.5.

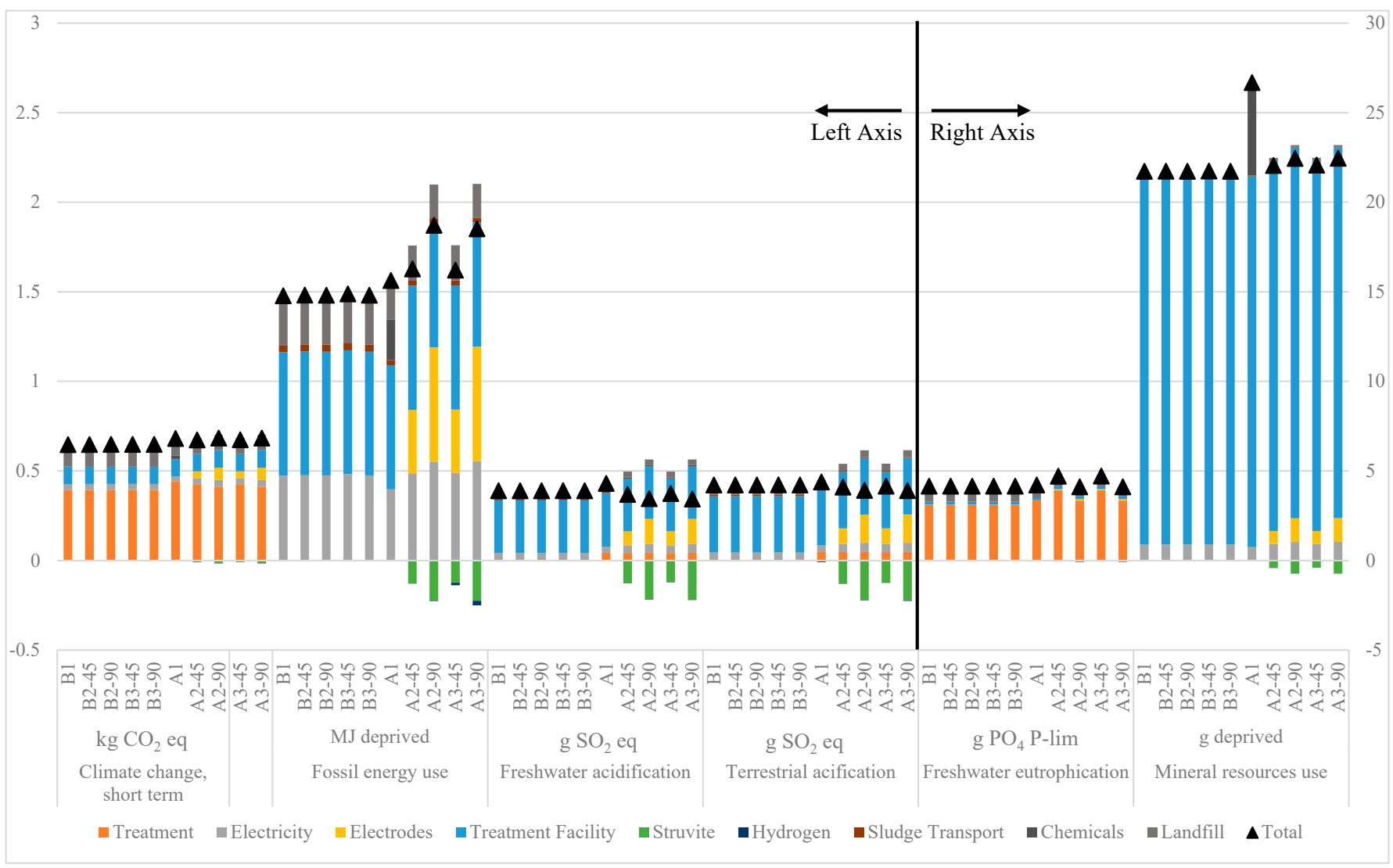

**Figure 5.** Process contribution results for wastewater treatment scenarios (6 categories).

### 3.2.1. Climate Change, Short Term

All B scenarios had lower relative impacts compared to A scenarios for climate change. Figure 5 shows the highest contributing group to this category for all scenarios are the emissions to air from the treatment process. The treatment facility contributed approximately 14–15% of the total climate change impact, while the electrodes and electricity each contributed equal to or less than 10% for each scenario. Landfilling of sludge contributed between 12–19% of emissions. Emissions to air from treatment contributed between 0.39–0.44 kg $CO_2$-eq. These emissions were higher for A scenarios than B scenarios. Based on GPS-X results, A scenarios seemed to have higher $N_2O$ emissions released per hour from the stabilization basins compared to the B scenarios. Release of nitrogen bound in cell mass from anaerobic digestion led to higher nitrate concentrations recirculating through the system, and therefore higher $N_2O$ emissions were released from the stabilization tanks [83]. These model results highlight a negative consequence in adding an anaerobic digester to a treatment process with regard to short term climate change. Additionally, adding the electrochemical struvite reactor for A scenarios produced higher $N_2O$ emissions compared to the A1 scenario due to electrode manufacturing. Credits received from struvite recovery offset burdens from electrode manufacturing.

### 3.2.2. Freshwater Eutrophication

Overall freshwater eutrophication impacts ranged between 4.1–4.7 g $PO_4$, with the A2-45 and A3-45 scenarios showing the highest overall impact and the A2-90 and A3-90 scenarios showing the lowest impact. Phosphorus emissions to water from the treatment process were the driving factors in aquatic eutrophication impacts for all scenarios. As mentioned in Section 3.1, addition of an anaerobic digester to the base case configuration resulted in a higher concentration of TP in the liquid effluent for all A scenarios. However, addition of electrochemical struvite recovery aided in recovering excess phosphorus resulting from anaerobic digestion. Total phosphorus in the sludge was reduced by 30–40% in A scenarios compared to B scenarios. Small differences in overall impacts between A2-45 and A3-45 as well as A2-90 and A3-90 show that inclusion of hydrogen generation had little effects in this impact category.

### 3.2.3. Terrestrial Acidification

Terrestrial acidification measures changes in chemical properties of soil due to atmospheric deposition of nutrients in acidifying forms including $NO_x$, $NH_3$, and $SO_2$. Overall impacts ranged between 0.39–0.44 g $SO_2$-eq with all B and A1 scenarios in this category showing higher overall impacts compared to the rest of the scenarios. The A1 scenario had the highest overall impacts, while the lowest overall impacts were achieved by the A3-90 scenario. The largest contributing group for all scenarios in this category was facility construction, contributing 0.31 g $SO_2$-eq primarily from $NO_x$ emissions from construction. In contrast to B scenarios, A scenarios also had larger contributions from the treatment process except the A1 scenario. These emissions from treatment were sourced from $NO_x$ emissions released by biogas combustion, highlighting a potentially significant drawback from biogas combustion to offset electricity requirements. A scenarios with electrochemical struvite recovery showed environmental credits ranging between 0.13–0.22 g $SO_2$-eq. However, these credits offset the impacts incurred by electrode manufacturing (0.08–0.15 g $SO_2$-eq).

### 3.2.4. Freshwater Acidification

A and B scenarios showed similar overall impacts for freshwater acidification, with overall impacts ranging between 0.34–0.43 g $SO_2$-eq across all scenarios. The largest contributor to this category was primarily the treatment facility. The A1 scenario showed the highest overall impacts resulting from higher impacts from chemicals coupled with no credits obtained from struvite and hydrogen gas. The A2-90 and A3-90 scenarios had the lowest impacts in this category resulting from struvite and hydrogen gas credits. The

A1 scenario showed 17% lower impacts from electricity compared to B scenarios for this category. Struvite production provided between 0.12–0.22 g $SO_2$-eq of environmental credits. However, similar to results from terrestrial acidification and nutrification, these credits mostly offset impacts from electrode manufacturing (0.07–0.14 g $SO_2$-eq).

### 3.2.5. Mineral Resources Use

Mineral resources use impacts ranged between 21–26 g deprived with facility construction making up approximately 20.7 g deprived for all scenarios. All A scenarios showed higher overall impacts compared to B scenarios by approximately 1–2 g deprived except A1 which showed almost 5 g deprived higher compared to B scenarios due to impacts from chemicals. Electricity was the second largest contributor this category, while electrode impacts were relatively small (0.7–1.3 g deprived). Struvite recovery provided credits between 0.42–0.73 g deprived, showing benefits from struvite recovery did not outweigh impacts incurred from electrode manufacturing in this impact category.

### 3.2.6. Fossil Energy Use

Overall impacts for non-renewable energy ranged between 1.5–1.9 MJ deprived per $m^3$ of treated wastewater. B3-90 achieved the lowest burdens in this category while the A2-90 showed the highest impacts. Similar to most impact categories, facility construction contributed the most to non-renewable energy impacts (0.69 MJ). Electricity (0.40–0.55 MJ) and electrodes (0.35–0.64 MJ) were also larger contributors. Struvite benefits ranged between 0.12–0.23 MJ, while hydrogen benefits were smaller between 0.001–0.025 MJ. Based on these results, offsets from hydrogen generation are generally small compared to overall impacts.

### 3.2.7. Other Impact Categories

Struvite credits did not completely offset the burdens from electrode manufacturing in the categories of human toxicity cancer, long-term climate change, land transformation, land occupation, photochemical oxidant formation, ionizing radiations, and particulate matter formation. Struvite credits were greater than electrode manufacturing burdens for the categories of freshwater ecotoxicity, ozone layer depletion and human toxicity non-cancer. Water scarcity was primarily driven by landfill emissions, showing lower emissions for A scenarios due to reductions in sludge that was landfilled. Marine eutrophication was primarily driven by the treatment process as well as landfill emissions. These results show that electrode manufacturing is a significant contributor to environmental impacts for this technology.

### *3.3. Economic Analysis and Results*
### 3.3.1. Cost and Revenue Analysis

For scenarios without anaerobic digestion (B scenarios), addition of struvite recovery and hydrogen capture show a minimal effect on costs, while also bringing negligible economic benefits (Table 6). Adding anaerobic digestion reduced capital costs associated with sludge hauling and landfilling by −23.5%, but this decrease was not enough to offset the rise in total capital costs for the A1 scenario. Lower sludge volumes reduced material costs by 11.3%, with biogas utilization decreasing energy costs by 15.6%. However, increases in capital costs plus added chemical and labor costs for struvite scale control using $FeCl_3$ resulted in a net cost increase of 6.9% for the A1 scenario. Anaerobic digestion increased soluble phosphate available for struvite precipitation, thus raising the potential for revenues to be generated from struvite sales across the A scenarios. With 45.0% phosphate capture (A2-45 & A3-45 scenarios) resulting annual revenues from the sale of struvite fertilizer were estimated at $163,627. With 90.0% phosphate capture (A2-90 & A3-90 scenarios) expected revenues rose to $295,438 per year. The capture of hydrogen from these systems brought further annual revenues of $12,624 and $22,793 for the A3-45 and A3-90 scenarios, respectively. Although struvite recovery and hydrogen capture showed potential for generating revenues, these were not large enough to offset accompanying increases in

capital and operating costs. At 45.0% phosphate recovery, net costs rose from $0.31/m$^3$ to $0.35/m^3$. At 90.0% phosphate recovery, these costs rose to $0.39/m^3$, a 22.7% increase from the baseline scenario.

**Table 6.** Annualized cost and revenue streams.

| Cost Category | B1 | B2-45 | B2-90 | B3-45 | B3-90 | A1 | A2-45 | A2-90 | A3-45 | A3-90 |
|---|---|---|---|---|---|---|---|---|---|---|
| Capital Costs | | | | | | | | | | |
| Standard Processes | $3,688,420 | $3,688,420 | $3,688,420 | $3,688,420 | $3,688,420 | $3,785,057 | $3,607,614 | $3,605,763 | $3,605,787 | $3,604,067 |
| Sludge Management | $60,357 | $60,357 | $60,357 | $60,357 | $60,357 | $46,189 | $44,797 | $44,797 | $44,797 | $44,797 |
| Anaerobic Digestion | $0 | $0 | $0 | $0 | $0 | $144,679 | $124,041 | $124,041 | $124,041 | $124,041 |
| ECST Reactor | $0 | $11 | $21 | $11 | $21 | $0 | $307,108 | $554,500 | $307,108 | $554,500 |
| H$_2$ Compression | $0 | $0 | $0 | $119 | $162 | $0 | $0 | $0 | $13,401 | $17,590 |
| H$_2$ Storage | $0 | $0 | $0 | $0 | $1 | $0 | $0 | $0 | $11,114 | $20,066 |
| Total Capital Cost | $3,748,777 | $3,748,788 | $3,748,798 | $3,748,907 | $3,748,961 | $3,975,925 | $4,083,560 | $4,329,101 | $4,106,248 | $4,365,062 |
| Operational Costs | | | | | | | | | | |
| O&M | $485,400 | $485,402 | $485,403 | $485,447 | $485,465 | $540,000 | $560,570 | $599,696 | $565,684 | $606,408 |
| Materials | $598,063 | $598,596 | $599,483 | $598,596 | $599,483 | $530,248 | $493,634 | $482,719 | $495,763 | $485,559 |
| Chemicals | $141,000 | $141,016 | $141,031 | $141,016 | $141,031 | $317,015 | $598,327 | $968,341 | $598,327 | $968,341 |
| Energy | $202,720 | $204,248 | $203,786 | $204,248 | $203,786 | $171,036 | $212,991 | $240,074 | $213,828 | $241,583 |
| Total Operational Cost | $1,427,183 | $1,429,261 | $1,429,704 | $1,429,306 | $1,429,766 | $1,558,299 | $1,865,522 | $2,290,829 | $1,873,603 | $2,301,891 |
| Potential Revenues | | | | | | | | | | |
| Struvite Fertilizer [a] | $0 | $6 | $11 | $6 | $11 | $0 | $163,627 | $295,438 | $163,627 | $295,438 |
| H$_2$ Gas [b] | $0 | $0 | $0 | $0 | $1 | $0 | $0 | $0 | $12,624 | $22,793 |
| Total Revenue | $0 | $6 | $11 | $6 | $12 | $0 | $163,627 | $295,438 | $176,251 | $318,230 |
| Net Cost | | | | | | | | | | |
| $/year | $5,175,960 | $5,178,043 | $5,178,491 | $5,178,207 | $5,178,715 | $5,534,224 | $5,785,455 | $6,324,493 | $5,803,600 | $6,348,722 |
| $/m$^3$ | $0.31 | $0.31 | $0.31 | $0.31 | $0.31 | $0.33 | $0.35 | $0.38 | $0.35 | $0.38 |

[a] Struvite sold at $1.15 per kg (the June 2022 price of diammonium phosphate). [b] Hydrogen gas sold at $8.00 per kg (the gasoline equivalent of $3.20 per gallon).

### 3.3.2. Break-Even Analysis of Struvite and Hydrogen Production

Capital costs for struvite production were estimated at $2.16/kg. These costs represent materials required for building and installing the struvite reactor and processing machinery, as well as the cost of building space to house the unit. O&M costs representing labor and materials and were estimated at $0.34/kg struvite. At a cost of $16.51/kg for magnesium metal (99.9%) and $0.06/kWh electricity, the value of Mg and energy consumed per kg of struvite produced equates to $3.24 and $0.29, respectively. To recover these capital and input costs, struvite fertilizers would need to be sold at a price of $6.03/kg (Table 7). Capital costs for hydrogen compression and storage were estimated at $13.22/kg H$_2$, assuming an H$_2$ production rate of 7.81 kg H$_2$ per day. With additional O&M and energy costs, the price needed to recover the costs of compression and storage was $15.58/kg H$_2$.

**Table 7.** Break-even price for struvite production and hydrogen capture.

| | Struvite [1] | Hydrogen [2] |
|---|---|---|
| | ($/kg) | ($/kg) |
| CAPEX | 2.16 | 13.22 |
| O&M | 0.34 | 2.36 |
| Magnesium Metal (99.9%) | 3.24 | - |
| Energy (electricity) | 0.29 | 0.01 |
| Break-Even Price | 6.03 | 15.58 |

[1] CAPEX represents costs for reactor materials/installation and building space. Energy based on electricity used for operating the struvite reactor. [2] CAPEX represents costs for compression and storage at flow rate of 7.81 kg H$_2$/day. Energy based on electricity required for H2 compression.

## 4. Discussion

### 4.1. Comparison of Results to Previous Studies

Resulting P recoveries of 18–33% based on influent P concentrations in this analysis are comparable to some recoveries reported in the literature for liquid stream-based P recovery technologies. Sena et al. reports a P recovery of 22% for chemical precipitation (Ostara) of struvite in liquid stream after dewatering of sludge [15]. Remy and Jossa report P recoveries between 7–12% for struvite recovery from digested sludge [10], while Amann et al. reports P recoveries of 10–60% (10–25% for Ostara) from liquid streams (aerobic centrate) [8]. It is important to note that Remy and Jossa did not include the entire WWTP in their LCI, but

rather just biosolids treatment and dewatering. All of these studies included an anaerobic digester in their system boundary for one or more scenarios modeled that allowed for nutrient release during biosolids treatment. Therefore, comparison of results to scenarios without an anaerobic digester is not possible.

Comparison of LCIA results to literature can be difficult as the system boundary, functional unit and impact method chosen for any study can largely affect impact results. Sena et al. provides the most comparable study to this study in that (1) the full WWTP with struvite recovery was modeled, (2) a functional unit of 1 m$^3$ was chosen, and (3) similar processes were included in the life cycle inventory with exception to construction of the treatment facility. They report a global warming potential of 0.31 kg $CO_2$-eq and acidification potential of 0.00262 kg $SO_2$-eq for 1 m$^3$ of treated water using the TRACI impact method [15]. Results of 0.65–0.68 kg $CO_2$-eq for short-term climate change impacts are approximately double that of Sena et al. For acidification, results in this study (compared to freshwater acidification) showed lower impacts (0.0003–0.0004 kg $SO_2$-eq). Sena et al. did not include facility construction in their LCI. As results show that infrastructure contributed significantly to environmental impacts across all categories, this may explain the differences in global warming impacts. Additionally, as electrochemical struvite recovery technology is not yet mature, economies of scale were not included in this analysis. For comparison of results in the eutrophication potential category, Rodriguez-Garcia et al. provides a similarly comparable study as Sena et al. with the exception of the use of a different functional unit for most of the study (kg $PO_4^{3-}$ eq removed) [9].They report approximately 8 g $PO_4^{3-}$ eq per m$^3$ of treated water for chemical precipitation of struvite placed post-treatment (anaerobic digester supernatant). This value is almost double the value of eutrophication potential estimated for electrochemical struvite precipitation in this study. This difference may be explained by the inclusion of sludge application to land rather than landfilling of sludge.

Previous studies have shown chemical struvite recovery to be economically feasible at relatively low flow capacities (265–3711 m$^3$/day) [84]. However, in a review of process economics for struvite crystallization Li et al. note that struvite crystallization is not likely to be economically feasible without a reduction in the cost of chemical inputs and/or a consideration of the monetary value of environmental benefits [13]. In this study, costs associated with struvite recovery alone were \$0.09/m$^3$ of treated wastewater and \$6.03 per kg of struvite produced. Sena et al. (2020) show a cost of \$0.09/m$^3$ for struvite recovery [85]. In their study they provide two values for struvite production. Based on these values, their costs would range from \$2.62/kg struvite to \$9.58/kg, putting results from this study within this range. Given the value of capital and input costs estimated for electrochemical struvite recovery in this study, the price of commercial fertilizers would need to increase by more than 400% for the process to become economically viable. Alternatively, a monetary value of \$10.41/kg $P_2O_5$ paid for avoided phosphates going to landfill would be sufficient to offset costs for electrochemical recovery. This value falls within ranges noted by Sena et al. from previous studies estimating the economic value of environmental benefits gained from P recovery [85].

*4.2. Future Directions*

Results showed that electrode manufacturing (particularly that of the Mg anode) is a hotspot in the environmental and economic performance of this novel technology. Thus, future research and design of electrochemical struvite recovery should focus on optimizing Mg consumption in struvite precipitation to efficiently utilize this resource. Additionally, hydrogen gas generation could help offset burdens. Theoretical $H_2$ yields used in this study were conservative, and results could be underestimating potential environmental and economic credits from hydrogen. Future experimental studies should measure actual $H_2$ yields and study how yields are affected by various design parameters.

Lastly, similar to Sena et al., addition of struvite technology as "side stream" or downstream process primarily affected nutrient recovery from the solid stream, not the liquid

effluent stream with regard to nutrient concentrations and associated eutrophication impacts. Similar changes in liquid effluent nutrient concentrations from struvite recovery were seen in both studies, highlighting an important aspect of nutrient recovery in wastewater treatment: potential freshwater eutrophication benefits from nutrient recovery in liquid effluents post-treatment will be limited.. Nutrient concentrations in liquid effluent cannot be decreased significantly with only side stream nutrient recovery downstream of the treatment train; rather, it is necessary to increase the overall treatment capacity of the plant (i.e., treatment parameters such as volume of anoxic/aerobic tanks). Future studies should focus on LCA of nutrient recovery technologies in WWT placed either (1) farther upstream in the treatment process, and/or (2) in a decentralized treatment scheme (before reaching the plant). As economic considerations can play a substantial role in dictating the adoption of treatment technologies, it would be beneficial to evaluate and incorporate the monetary environmental benefit of nutrient pollution reduced by recovery technologies in future research.

## 5. Conclusions

New technological advances have created opportunities for redesigning conventional wastewater treatment plants to not only include treatment of waste, but also recover valuable nutrients necessary for life. While novel technology such as electrochemical struvite recovery is still under research and development, prospective life cycle assessment and economic analysis can shed light on important potential "hotspots" in its performance in a simulated real-world environment prior to its integration into a pilot plant. Results from this study show electrochemical struvite recovery and hydrogen capture appear to offer some benefits in the form of environmental credits and revenues generated through the sale of captured outputs. Environmental benefits include reductions in freshwater eutrophication and terrestrial acidification. Economic benefits include revenue generated from struvite and hydrogen gas. However, under current operating conditions, costs outweigh these benefits, with magnesium inputs being the largest limiting factor in both environmental and economic feasibility of the process. Future research and development of this technology should focus on this input to increase economic feasibility in addition to designing for the environment.

**Supplementary Materials:** The following supporting information can be downloaded at: https://www.mdpi.com/article/10.3390/su142013657/s1.

**Author Contributions:** Conceptualization, K.G.M. and L.E.; methodology, K.G.M. and L.E.; software, K.G.M. and L.E.; validation, K.G.M., L.E., G.T. and J.P.; formal analysis, K.G.M., L.E., G.T. and J.P.; investigation, K.G.M. and L.E.; resources, G.T. and J.P.; data curation, K.G.M. and L.E.; writing—original draft preparation, K.G.M. and L.E.; writing—review and editing, G.T. and J.P.; visualization, K.G.M. and L.E.; supervision, G.T. and J.P.; project administration, K.G.M. and L.E.; funding acquisition, G.T. and J.P. All authors have read and agreed to the published version of the manuscript.

**Funding:** This research was funded by the NSF Innovations at the Nexus of Food, Energy and Water Systems Award Abstract #1739473. The APC was also funded by the NSF INFEWS Award.

**Institutional Review Board Statement:** Not applicable.

**Informed Consent Statement:** Not applicable.

**Data Availability Statement:** Not applicable.

**Acknowledgments:** We would like to thank Lance McAvoy, Johnny Key and the Ft. Smith P Street operators for their valuable help in providing data for the treatment plant. We would also like to thank Kamyar Sardari for his help in building and troubleshooting our base case model in GPS-X.

**Conflicts of Interest:** The authors declare no conflict of interest. The funders had no role in the design of the study; in the collection, analyses, or interpretation of data; in the writing of the manuscript, or in the decision to publish the results.

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
