# Peer review of "Prospective Life Cycle Assessment and Cost Analysis of Novel Electrochemical Struvite Recovery in a U.S. Wastewater Treatment Plant"

_sustainability, doi:10.3390/su142013657_

Round 1

Reviewer 1 Report

Dear authors, please find bellow suggestions and comments for your work.

As you are talking about novel technology, why didnt you built small pilot WWTP to avoid certain assumptions and then use models?

Row 118: Please put informations about WWTP: what is design flow and hom much is the population equivalent (PE)?

Table 1: Please write correct units and symbols (sub and super script...)

Row 135: Please better explain "Black Box 1" in the manuscript, i.e. processes and balanced chemical reaction equation.

Row 137: How do you assume struvite yields in this novel technology?

Expand conclusions based on results/discussion.

When do you plan to use this technology on describetd WWTP?

It is necessary to work on the very structure of the work and try to shorten it. In part 2, it is necessary to explain the selected scenarios as best as possible and to focus on newer technology. It is necessary to explain certain assumptions in the work and expand the conclusions.

It is recommended to use the passive in forming sentences, not in the first person singular or plural.

Reviewer 2 Report

In this paper, a prospective life cycle assessment and cost analysis of a new electrochemical struvite recovery technology that utilizes a sacrificial magnesium anode to precipitate struvite and generate hydrogen gas were conducted. The technology was modeled using process simulation software GPS-X and CapdetWorks assuming its integration in a full-scale existing wastewater treatment plant with and without anaerobic digestion. It is an interesting content, but arranged structure needs to be further improved. Therefore, it needs minor revision before it is published in this journal. Some issues should be carefully addressed. Please check the attachment.

Reviewer 3 Report

The manuscript “Prospective Life Cycle Assessment and Cost Analysis of Novel Electrochemical Struvite Recovery in a U.S. Wastewater Treatment Plant”, by Karla G. Morrissey, Leah English, Greg Thoma, and Jennie Popp, presents results on the comparison of different strategies for domestic wastewater treatment with the incorporation of phosphorous recovery through the (electrochemical) production of struvite. A plant currently in operation was considered as a reference and, taking into account the desired products (struvite and/or hydrogen gas), a life cycle inventory was performed. The advantages/disadvantages of the treatments (from an environmental point of view) were assessed through the comparison of outputs for each impact category. The economic analysis was a complement to the application of the LCA and, it is expected, should support conclusions.

In general, the manuscript is easy to understand and follows a logical sequence. The introduction is clear and the methodology (materials and methods) provides enough information on the work. I would suggest the authors include in the introduction the industrial uses of struvite since this allows visualizing the commercial importance of this product resulting from the wastewater treatment. The basic information of the life-cycle assessment, or LCA, is provided by the authors. The functional unit as well as the employed software, method and scenarios were properly presented.

I suggest improvements in the format and presentation of the Supplemental Material. The modified code should be presented in a more understandable form, i.e., with a title and a short presentation of the information. The Process Flow Diagrams are presented and this is very useful for the reader. However, larger images of these flow diagrams should be presented; the authors could present one flow diagram on a single page, with a legend: “Figure S1. Process Flow Diagram of…”.

Concerning the results and discussion, the authors focused on the comparison of the impacts calculated for each impact category. I wonder why the authors did not calculate the (mili)points to (numerically) compare the treatments. Probably, with these results, they could propose the most environment-friendly treatment. In fact, a conclusion referred to the LCA is missing.

I have included some comments in the PDF document.

In summary, this is a well-written paper and the use of results, for interpretation purposes, was carefully planned. Some minor aspects should be clarified to improve the manuscript. After minor revisions, this paper could be considered for publication in Sustainability.

Round 2

Reviewer 1 Report

Dear authors.

I have only two comments, and that is to include sentence from your first comment/response to me, in the conclusions. Just rewrite it so that can be incorporated in the manuscript, part conclusions. You write: 

Response: Our technology is bench-scale and is currently under research and development. We conducted this analysis in order to simulate, to the best of our ability, how this technology would perform in a real world setting to get an idea of its environmental and economic performance and how it affects N and P fates in the treatment plant. Building a pilot plant was outside the scope of our analysis and funding was unavailable.

And second to correct writing chemical compounds:

Table 1: PO4-P (4 goes to subscript)

g CaCO3/m3

g O2/m3

Figures 4 and 5: kg CO2 ... kg SO2  ... kg PO4-P ...
